# Phosphorylation-mediated PTEN conformational closure and deactivation revealed with protein semisynthesis

**David Bolduc[1], Meghdad Rahdar[2], Becky Tu-Sekine[3], Sindhu Carmen Sivakumaren[1], Daniel Raben[3], L Mario Amzel[4], Peter Devreotes[5], Sandra B Gabelli[4,6]\*, Philip Cole[1]\***

[1]Department of Pharmacology and Molecular Sciences, Johns Hopkins University School of Medicine, Baltimore, United States; [2]Department of Pharmacology, University of California, San Diego, San Diego, United States; [3]Department of Biological Chemistry, Johns Hopkins University School of Medicine, Baltimore, United States; [4]Department of Biophysics and Biophysical Chemistry, Johns Hopkins University School of Medicine, Baltimore, United States; [5]Department of Cell Biology, Johns Hopkins University School of Medicine, Baltimore, United States; [6]Departments of Medicine and Oncology, Johns Hopkins University School of Medicine, Baltimore, United States

**Abstract** The tumor suppressor $PIP_3$ phosphatase PTEN is phosphorylated on four clustered Ser/Thr on its C-terminal tail (aa 380–385) and these phosphorylations are proposed to induce a reduction in PTEN's plasma membrane recruitment. How these phosphorylations affect the structure and enzymatic function of PTEN is poorly understood. To gain insight into the mechanistic basis of PTEN regulation by phosphorylation, we generated semisynthetic site-specifically tetra-phosphorylated PTEN using expressed protein ligation. By employing a combination of biophysical and enzymatic approaches, we have found that purified tail-phosphorylated PTEN relative to its unphosphorylated counterpart shows reduced catalytic activity and membrane affinity and undergoes conformational compaction likely involving an intramolecular interaction between its C-tail and the C2 domain. Our results suggest that there is a competition between membrane phospholipids and PTEN phospho-tail for binding to the C2 domain. These findings reveal a key aspect of PTEN's regulation and suggest pharmacologic approaches for direct PTEN activation.

**\*For correspondence:** gabelli@jhmi.edu (SBG); pcole@jhmi.edu (PC)

## Introduction

PTEN (phosphatase and tensin homolog deleted on chromosome 10) suppresses cell proliferation, migration and survival by dephosphorylating the lipid second messenger phosphatidyl inositol 3,4,5-triphosphate ($PIP_3$), thereby opposing phosphatidyl inositol-3 kinase (PI3K) signaling and preventing AKT protein kinase activation (*Maehama and Dixon, 1998*; *Myers et al., 1998*; *Sun et al., 1999*; *Iijima and Devreotes, 2002*). PTEN is one of the most frequently mutated genes in cancer, with inactivating mutations found in many solid tumor types (*Li et al., 1997*; *Steck et al., 1997*). The PTEN gene has been shown to be inactivated through somatic and germline mutations as well as transcriptionally suppressed through epigenetic mechanisms (*Salvesen et al., 2001*; *Meng et al., 2007*; *Hollander et al., 2011*). Homozygous deletion of PTEN is embryonically lethal in mice while heterozygous mice are predisposed to developing tumors (*Suzuki et al., 1998*; *Di Cristofano et al., 1999*; *Podsypanina et al., 1999*). Germline mutations in PTEN predispose people to spontaneous tumor formation as seen in Cowden syndrome (*Nelen et al., 1997*).

PTEN is a 47 kDa (403 aa) protein composed of a 'dual-specificity' phosphatase domain, a C2 domain that mediates membrane association, and a 52 aa C-terminal tail (*Figure 1A*). An X-ray

**eLife digest** PTEN is an enzyme that is found in almost every tissue in the body, and its job is to stop cells dividing. If it fails to perform this job, the uncontrolled proliferation of cells can lead to the growth of tumors. PTEN stops cells dividing by localizing at the plasma membrane of a cell and removing a phosphate group from a lipid called PIP$_3$: this sends a signal, via the PI3K pathway, that suppresses the replication and survival of cells.

Three regions of PTEN are thought to be central to its biological functions: one of these regions, the phosphatase domain, is directly responsible for removing a phosphate group from the lipid PIP$_3$; a second region, called the C2 domain, is known to be critical for PTEN binding to the cell membrane; however, the role of third region, called the C-terminal domain, is poorly understood.

Many proteins are regulated by the addition and removal of phosphate groups, and PTEN is no exception. In particular, it seems as if the addition of phosphate groups to four amino acid residues in the C-terminal domain can switch off the activity of PTEN, but the details of this process have been elusive.

Now, Bolduc et al. have employed a variety of biochemical and biophysical techniques to explore this process, finding that the addition of the phosphate groups reduced PTEN's affinity for the plasma membrane. At the same time, interactions between the C-terminal and C2 domains of the PTEN cause the shape of the enzyme to change in a way that 'buries' the residues to which the phosphate groups have been added.

In addition to offering new insights into PTEN, the work of Bolduc et al. could help efforts to identify compounds with clinical anti-cancer potential.

structure of PTEN, lacking the apparently flexible N-terminus (aa 1–13), internal D-loop (aa 286–309) and C-tail (aa 353–403), shows tight-knit interactions between the catalytic and C2 domains (*Lee et al., 1999*). The PTEN protein is believed to be regulated by a variety of mechanisms including post-translational modifications, protein–protein interactions, and protein–lipid interactions (*Campbell et al., 2003*; *Iijima et al., 2004*; *Walker et al., 2004*; *Vazquez et al., 2006*; *Denning et al., 2007*; *Chagpar et al., 2009*; *Shenoy et al., 2012*; *Song et al., 2012*). Previously mapped PTEN post-translational modifications include phosphorylation (*Vazquez et al., 2000*; *Torres and Pulido, 2001*; *Miller et al., 2002*; *Al-Kouri et al., 2005*; *Cordier et al., 2012*), acetylation (*Okumura et al., 2006*), ubiquitylation (*Wang et al., 2007*), and sumoylation (*Huang et al., 2012*). Phosphorylation of a cluster of Ser/Thr residues (Ser380, Thr382, Thr383, and Ser385) on PTEN's C-terminal tail have received considerable attention in studies on PTEN regulation and appears to be a major modification (*Wu et al., 2000*; *Vazquez et al., 2001*; *Odriozola et al., 2007*; *Rahdar et al., 2009*). Reported to be catalyzed by the protein kinases CK2 and/or GSK3β (*Torres and Pulido, 2001*; *Al-Khouri et al., 2005*), this phospho-cluster is mutationally sensitive in cell assays, with replacements by Ala driving PTEN to the plasma membrane (*Rahdar et al., 2009*). While the molecular mechanism of how this is achieved is uncertain, altered protein–protein interactions (*Wu et al., 2000*; *Vazquez et al., 2001*; *Sumitomo et al., 2004*; *Takahashi et al., 2006*; *van Diepen et al., 2009*; *Huang et al., 2012*) as well as conformational changes (*Odriozola et al., 2007*; *Rahdar et al., 2009*) in PTEN have been suggested to be induced by phosphorylation. It has been unclear if phosphorylation of PTEN has a direct effect on membrane association or is indirectly mediated through other macromolecular interactions. Most PTEN phosphorylation analyses have relied on cellular transfection experiments in which replacement of the Ser and Thr residues with Ala have been indirect indicators of phosphorylation function. However, Ser and Thr have different properties from Ala and such mutations may be misleading in deducing the action of a phosphoSer/phosphoThr (*Vazquez et al., 2000*; *Torres and Pulido, 2001*; *Al-Khouri et al., 2005*). What has been lacking thus far is a biochemical analysis of purified phosphorylated PTEN in which the phosphorylation sites are well-defined in position and stoichiometry. Inherent challenges in obtaining this material include the difficulty in using kinases for introducing phosphates site-specifically into PTEN and the proposed potential for PTEN autodephosphorylation (*Zhang et al., 2012*). To circumvent these issues, we employ here expressed protein ligation, a method for protein semisynthesis (*Schwarzer and Cole, 2005*), for generating 380,382,383,385-tetraphosphorylated-PTEN (4p-PTEN). Expressed protein ligation involves the production of a recombinant protein carrying a C-terminal thioester by exploiting the action of an intein, and its chemoselective ligation to an N-Cys

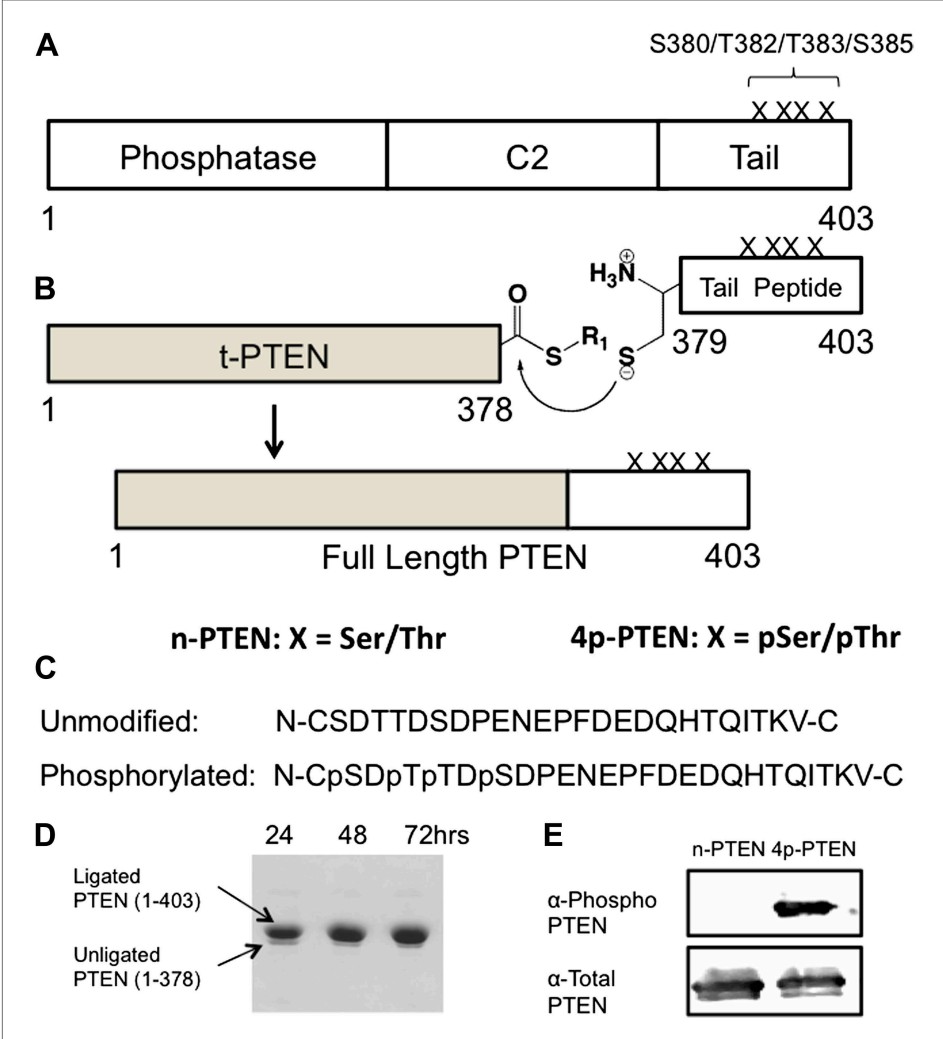

Figure 1. Generation of semisynthetic PTEN proteins. (**A**) PTEN is composed of phosphatase domain, a C2 domain, and a C-terminal tail that is phosphorylated multiple times within a cluster of Ser and Thr residues (S380/T382/T383/S385). (**B**) C-terminally truncated PTEN containing an intein generated thioester at its C-terminus is ligated to a synthetic PTEN tail peptide with or without phosphorylation at the S380/T382/T383/S385 cluster. The final product is full length PTEN in the phosphorylated (4p-PTEN) or unphosphorylated (n-PTEN) form. (**C**) C-terminal tail peptides were synthesized in the unphosphorylated form or phosphorylated at S380/T382/T383/S385. Note that the N-Cys replaces a natural Tyr in PTEN. (**D**) The ligation reaction precedes smoothly over 72 hr. (**E**) Western blot with an anti-phospho PTEN antibody reveals 4p-PTEN but not n-PTEN is phosphorylated.

The following figure supplements are available for figure 1:

**Figure supplement 1**. Schematic views of n-PTEN, 4p-PTEN and t-PTEN.

**Figure supplement 2**. MALDI spectra for PTEN tail peptides and semisynthetic PTEN proteins.

**Figure supplement 3**. Size exclusion chromatography, Y379C enzymatic characterization and autophosphatase activity of PTEN.

containing synthetic peptide (*Muir et al., 1998*; *Vila-Perelló and Muir, 2010*). Within the synthetic peptide, phosphoSer/phosphoThr can be installed using standard chemical techniques at specific locations. We show here that, relative to the unphosphorylated form, 4p-PTEN adopts a more compact conformation in which the C-tail appears to interact with the C2 domain in a fashion that reduces its affinity for lipid membranes and diminishes its catalytic activity.

## Results

### Semisynthesis of tetraphosphorylated and unphosphorylated PTEN (4p-PTEN and n-PTEN)

A requirement for expressed protein ligation is the correct positioning of a Cys for the native chemical ligation reaction (*Muir et al., 1998*; *Schwarzer and Cole, 2005*; *Vila-Perelló and Muir, 2010*). Initial attempts at generating semisynthetic PTEN focused on an *Escherichia coli* expression system. In this way, we showed that PTEN-intein fusion proteins allowed for generation of catalytically active recombinant PTEN thioester fragments (data not shown). We also determined using GST-PTEN that Cys was well-tolerated at the position needed for ligation, showing no change in catalytic activity induced by Y379C mutation (*Figure 1—figure supplement 3D*). However, expression in *E. coli* of the intein-fusion protein suffered from low yields of soluble PTEN protein expression (<0.1 mg/l culture) which was insufficient for our needs.

We thus subcloned aa 1–378 of PTEN (t-PTEN) into a baculovirus plasmid for insect cell expression. Ligation with a tetraphosphorylated (and unphosphorylated as a control) N-Cys synthetic peptide aa 379–403 (*Figure 1C*) proceeded smoothly over 72 hr, providing 8–10 mg/l of culture purified semisynthetic PTEN protein (*Figure 1B,D*). Tetraphosphorylated (4p-PTEN), unphosphorylated (n-PTEN) and C-terminally truncated PTEN (t-PTEN) were generated in this way (*Figure 1—figure supplement 1*). Semisynthetic proteins were >90% pure using Coomassie staining and their structural integrity confirmed by mass spectrometry (*Figure 1—figure supplement 2C and D*). Western blot with commercial anti-phospho-PTEN Ab showed that 4p-PTEN, and not n-PTEN, was appropriately phosphorylated (*Figure 1E*). It should also be noted that baculovirus systems have rarely been used for intein expression vectors (*Pradhan et al., 1999*), possibly in part because of the expression of chitinase in these hosts which interferes with the standard chitin affinity purification scheme. Nevertheless, we found the presence of chitinase to be a surmountable issue by a chromatographic pre-clearing step with a bed of fibrous cellulose. We also found that High Five insect cells rather than the more typical SF9 insect cells were critical for robust expression. Despite the theoretical concern about autodephosphorylation (*Zhang et al., 2012*), 4p-PTEN prepared in the presence of a PTEN phosphatase vanadate-based inhibitor was equally phosphorylated to that prepared in its absence (*Figure 1—figure supplement 3E*). Further experiments showed that 4p-PTEN did not undergo spontaneous pSer/pThr hydrolysis over 24 hr as monitored by western blot (*Figure 1—figure supplement 3F*).

### Enzymatic characterization of 4p-PTEN

Initial analysis of 4p-PTEN catalytic activity was carried out with a soluble $PIP_3$ substrate containing hexanoyl rather than the more physiological palmitoyl chains using a phosphate release detection assay with malachite green (*Van Veldhoven and Mannaert, 1987*). These studies revealed a ~sixfold reduction in catalytic efficiency ($k_{cat}/K_m$) conferred by C-terminal phosphorylation in which 4p-PTEN shows a significantly higher soluble $PIP_3$ $K_m$ (*Figure 2A*). To determine the enzymatic activity with a more physiologically relevant substrate, we analyzed 4p-PTEN's dephosphorylation of vesicle-incorporated $PIP_3$ (containing palmitoyl chains) (*McConnachie et al., 2003*). 3'-[$^{32}$P] $PIP_3$ was prepared by PI3-kinase and incorporated into vesicles containing unlabeled $PIP_3$ and phosphatidylcholine (PC) (*McConnachie et al., 2003*). For interfacial enzymes such as PTEN, the bulk concentration as well as the surface concentration of the substrate in the lipid bilayer needs to be considered (*Deems et al., 1975*; *Hendrickson and Dennis, 1984*; *McConnachie et al., 2003*). We thus explored the rate of hydrolysis of $PIP_3$ under conditions where the surface concentration of $PIP_3$ was held constant while varying the bulk concentration of $PIP_3$ and carrier lipid (phosphatidylcholine, PC) proportionately (bulk dilution). In addition, we measured PTEN activity under conditions where the surface concentration of $PIP_3$ was varied while its bulk concentration was held constant by varying the carrier lipid PC (surface dilution). As described in the 'Materials and methods' section, apparent $V_{max}$ and apparent $K_m$ values obtained from these experiments were fit to the equations in the 'Materials and methods' section associated with the equation below:

$$V_0 = (V_{max} * X_s * [S_0]) / (iK_m * K_s + iK_m * [S_0] + X_s * [S_0])$$

(*McConnachie et al., 2003*) in which $X_s$ is the surface concentration (mol fraction) of the substrate $PIP_3$, $S_0$ is the bulk concentration of $PIP_3$, $iK_m$ is the interfacial Michaelis constant (mol%) and $K_s$ is the

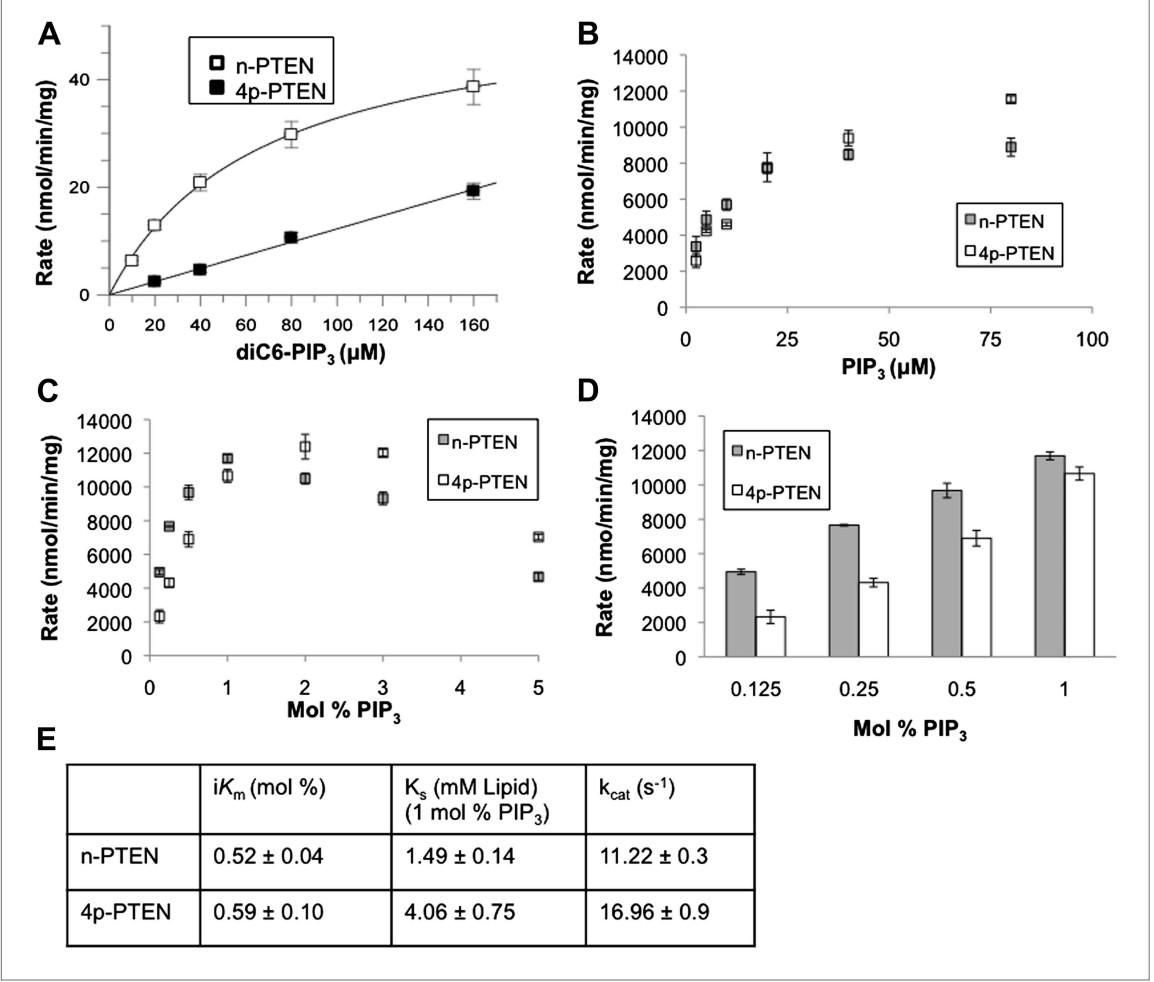

**Figure 2**. Soluble substrate activity and Interfacial kinetic analysis of semisynthetic PTEN. (**A**) PTEN activity to a soluble substrate, diC6-PIP$_3$. (n-PTEN: $k_{cat}$ = 2.6 ± 0.1 min$^{-1}$, $K_m$ = 67 ± 4.2 μM, $k_{cat}/K_m$ = 0.038 ± 0.001 min$^{-1}$μM$^{-1}$; 4p-PTEN: $k_{cat}/K_m$ = 0.005 ± 0.0002 min$^{-1}$μM$^{-1}$) (**B** and **C**) PTEN activity to palmitoyl PIP$_3$ incorporated into phosphatidylcholine vesicles. In the bulk dilution experiment (**B**) enzymatic activity for n-PTEN and 4p-PTEN was measured at a fixed surface concentration of 1% PIP$_3$ while the bulk concentration was varied. In the surface dilution experiment (**C**) activity was measured at a fixed bulk concentration of 50 μM PIP$_3$ while the surface concentration was varied. (**D**) 4p-PTEN has lower activity than n-PTEN only at low PIP$_3$ concentrations. (**E**) Summary of the interfacial kinetic analysis of n-PTEN and 4p-PTEN. Data are reported as the mean ± the SEM from three experiments performed in duplicate. Apparent V$_{max}$ values were obtained from the best fit curves from the first four points of the surface dilution experiments.

The following figure supplements are available for figure 2:

**Figure supplement 1**. Interfacial kinetic analysis of semisynthetic PTENs.

**Figure supplement 2**. Bulk and surface dilution curves of t-PTEN.

**Figure supplement 3**. Anionic lipid stimulation of n-PTEN and 4p-PTEN.

membrane dissociation constant for PTEN interaction with vesicles. Bulk dilution experiments yielded a rectangular hyperbola (**Figure 2B**), while surface dilution experiments showed apparent substrate inhibition at higher surface concentrations of PIP$_3$ (**Figure 2C**). Using the lower substrate concentrations where substrate inhibition was minimal, the interfacial kinetic analysis revealed that n-PTEN and 4p-PTEN have similar i$K_m$ values with minor differences in k$_{cat}$ values (**Figure 2E** and **Figure 2—figure supplement 1C**). However, at 1% PIP$_3$, the K$_s$ value of 4p-PTEN showed a ~threefold increase compared to that of n-PTEN, suggesting a decrease in binding affinity for the vesicle membrane

when PTEN is phosphorylated (*Figure 2E*). n-PTEN $K_s$ and i$K_m$ values were similar to those of t-PTEN (*Figure 2—figure supplement 2*), indicating that the unphosphorylated tail extension does not hinder membrane interactions. Our data suggest that at low (<1 mol%) PIP$_3$ concentrations which are physiologic, the rate differential between n-PTEN and 4p-PTEN is significant (*Figure 2D*).

## Anionic lipid vesicle 4p–PTEN-interactions

To further analyze the apparent membrane affinity loss conferred by PTEN tail phosphorylation, we explored the binding interactions of semisynthetic PTENs with vesicles containing anionic phospholipids. As reported previously for unphosphorylated PTEN produced in *E. coli* (*McConnachie et al., 2003*), we found that increasing levels of both phosphatidylserine (PS) and PIP$_2$ in vesicles positively influence PIP$_3$ dephosphorylation by both n-PTEN and 4p-PTEN (*Figure 2—figure Supplement 3*). We next carried out a series of PTEN-vesicle binding assays using PIP$_2$ and PS as membrane targeting lipids by measuring differential sedimentation fractionation. These experiments demonstrated markedly reduced membrane binding for 4p-PTEN vs n-PTEN under all conditions tested (*Figure 3A,B* and *Figure 3—figure supplement 1*). Both n-PTEN and 4p-PTEN show proportionally greater binding to vesicles as concentrations of PIP$_2$ and PS are increased, but 4p-PTEN compared with n-PTEN requires a significantly higher concentration of anionic lipid to achieve a similar level of membrane sedimentation. For example, 7 mol% PIP$_2$ sedimented about one-fifth of 4p-PTEN, which was comparable to the amount of n-PTEN sedimented by 3 mol% PIP$_2$ (*Figure 3A*). With 5 mol% PS, the ratio of n-PTEN:4p-PTEN sedimented was about 10:1 (*Figure 3B*). These results corroborate the PTEN affinity differences observed in the phosphatase assays where C-tail phosphorylation of PTEN inhibited membrane binding.

## Conformational analysis of 4p-PTEN using ion exchange chromatography, limited proteolysis, and alkaline phosphatase sensitivity

It has been proposed that C-terminal phosphorylation of PTEN may alter its conformation (*Odriozola et al., 2007*; *Rahdar et al., 2009*). In the course of purification of our semisynthetic PTENs by anion exchange chromatography, we made the paradoxical observation that the tetraphosphorylated PTEN reproducibly eluted at an earlier elution volume (~70 ml and lower salt concentration, 90 mM NaCl) and more sharply relative to the unphosphorylated protein (~100 ml, 140 mM NaCl) (*Figure 4A*). Elution of n-PTEN as one large peak followed by multiple small peaks was attributed to minor heterogeneous insect cell-mediated phosphorylation in the recombinant tail moiety on and near Thr366 and Ser370 as detected with a site-specific Ab (*Figure 4—figure supplement 1B*). A similar elution pattern was observed with t-PTEN (*Figure 4—figure supplement 1A*). This heterogeneity is presumably collapsed under one peak in 4p-PTEN. Since 4p-PTEN has a nominal eight negative charge increase relative to n-PTEN, we were surprised that 4p-PTEN showed decreased affinity to cationic resin. One explanation for this result is that, relative to n-PTEN, 4p-PTEN undergoes a conformational change which buries its negatively charged C-tail, reducing its availability for interacting with cationic resin. The fact that clusters of charges can be more important than

**Figure 3**. n-PTEN and 4p-PTEN binding to large multilamellar vesicles (LMVs). The percent of n-PTEN and 4p-PTEN bound to sedimented phosphatidylcholine LMVs incorporated with (**A**) PIP$_2$ or (**B**) phosphatidylserine was determined by quantification of western blot bands. Data are reported as the mean ± the SEM of three separate experiments.

The following figure supplements are available for figure 3:

**Figure supplement 1**. n-PTEN and 4p-PTEN binding to large multilamellar vesicles (LMVs).

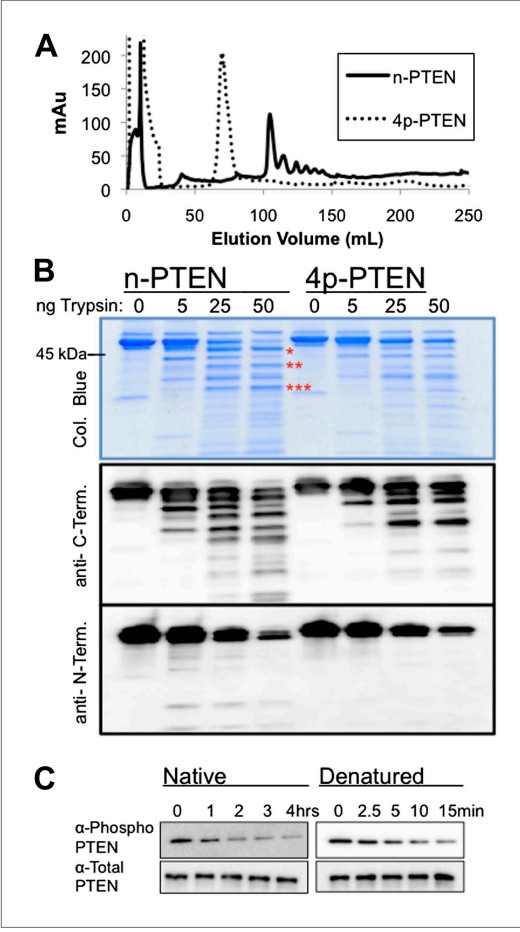

Figure 4. Conformational changes associated with PTEN phosphorylation. (**A**) With a gradient of 0–50% NaCl over 250 ml on an anion exchange column, 4p-PTEN elutes at ~70 ml while n-PTEN elutes at ~100 ml. (**B**) 2 µg of n-PTEN and 4p-PTEN were digested with varying amounts of trypsin then visualized by colloidal blue staining or by western blot with an antibody to the N- or C-terminus of PTEN. Asterisks denote bands that are in higher abundance in the digestion of n-PTEN compared to 4p-PTEN. N-terminal sequencing of these bands identifies the cleavage sites as (\*) R15, (\*\*) R84 and (\*\*\*) R161. (**C**) Denatured 4p-PTEN treated with 0.5 µM alkaline phosphatase is significantly more sensitive to dephosphorylation of the tail phosphocluster compared to the native form of 4p-PTEN treated with 1 µM alkaline phosphatase.

The following figure supplements are available for figure 4:

**Figure supplement 1**. Non-tail cluster phosphorylation of PTEN expressed in High Five insect cells.

**Figure supplement 2**. Native and denatured 4p-PTEN sensitivity to alkaline phosphatase.

overall charge for protein–ion exchange resin interactions has been discussed previously (*Chung et al., 1989*; *Hou et al., 2010*).

To further explore the conformation of 4p-PTEN, we compared its susceptibility to trypsin proteolysis relative to n-PTEN. 4p-PTEN appeared more protease resistant vs its unphosphorylated counterpart as seen most clearly at 25 and 50 ng trypsin (*Figure 4B*). Based on western blots with N-terminal and C-terminal PTEN Abs, several large metastable fragments still containing C-terminal epitopes were observed in n-PTEN, indicative of enhanced protease sensitivity in the catalytic domain in the unphosphorylated protein. N-terminal sequencing identified several of these cleavage positions (*Figure 4B*).

Alkaline phosphatase sensitivity and western blotting was used to assess the exposure of the C-terminal 380–385 phospho cluster in 4p-PTEN (*Wang et al., 2002*). The half-life of alkaline phosphatase-mediated dephosphorylation of 4p-PTEN under the conditions used was found to be ~60 min (*Figure 4C* and *Figure 4—figure supplement 2A*). Serendipitously, we found that freeze-thawing dilute 4p-PTEN led to apparent denaturation of the protein since it was much more susceptible to alkaline phosphatase catalyzed dephosphorylation of the tail phosphate cluster. Under the same conditions, near complete dephosphorylation was observed within 7.5 min. To more precisely determine this rate, we cut in half the concentration of alkaline phosphatase, and assumed that this process follows a pseudo-first-order kinetic mechanism. In this way, we estimate that native 4p-PTEN vs denatured 4p-PTEN shows a 25-fold reduced rate of alkaline phosphatase-catalyzed dephosphorylation of the tail phospho cluster (*Figure 4C* and *Figure 4—figure supplement 2B*). We infer that this rate-differential suggests a closed:open 4p-PTEN conformational equilibrium of 25:1 under the conditions of this experiment.

## Small angle X-ray scattering (SAXS) analysis of semisynthetic PTENs

To gain more information on the structural changes induced by PTEN tail phosphorylation, we performed small angle X-ray scattering (SAXS) analysis of t-PTEN, n-PTEN, and 4p-PTEN. The scattering plots are shown in *Figure 5A* for each protein. The shape of the pair distance distribution function p(r) plot for n-PTEN reveals a single hump with a large shoulder region at higher r values (*Figure 5B*). The shoulder region is indicative of a protein with an elongated shape (*Putnam et al., 2007*; *Jacques and Trewhella, 2012*), possibly due to the presence of the extended tail. The shoulder is reduced for t-PTEN and 4p-PTEN, suggesting the tail is no longer in an extended position (*Figure 5B*).

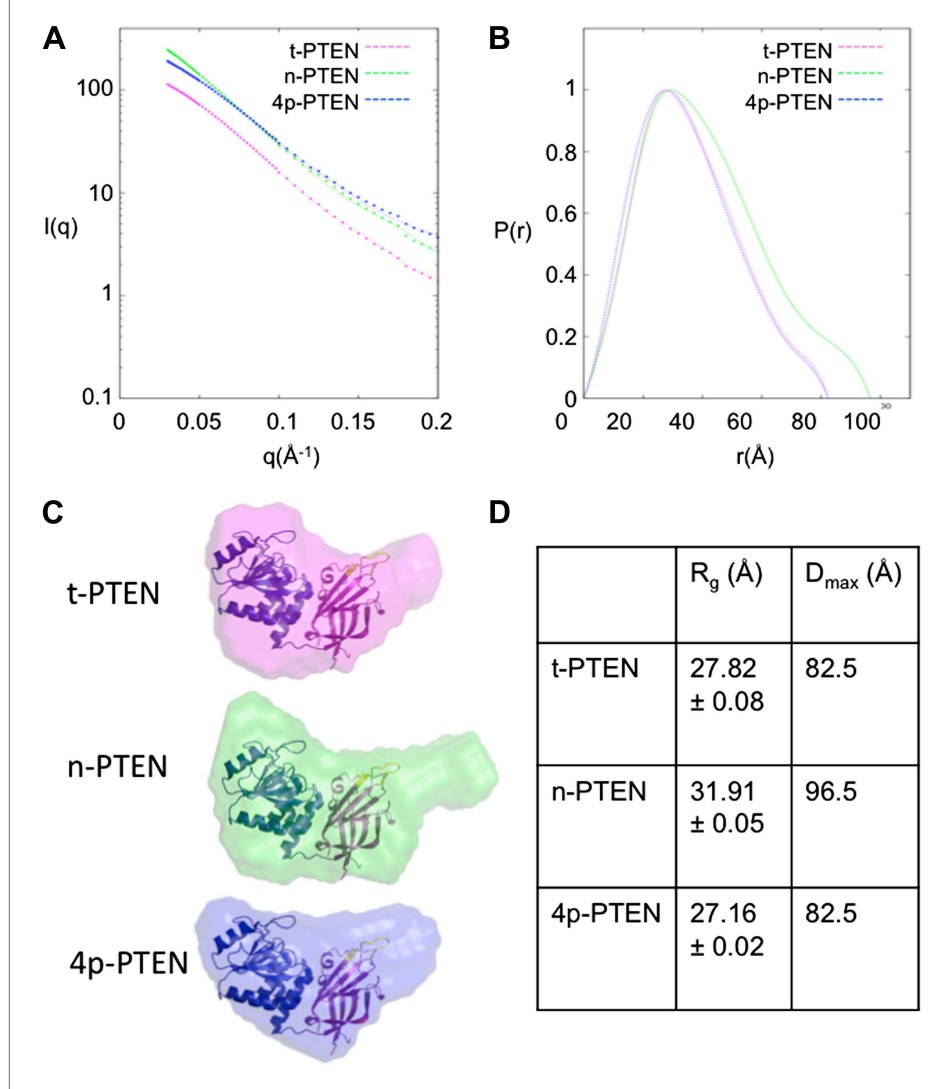

**Figure 5**. SAXS analysis for t-PTEN, n-PTEN and 4p-PTEN. (**A**) Scattering diagrams and (**B**) pair distribution function (Pofr) plots for t-PTEN, n-PTEN, and 4p-PTEN. (**C**) The molecular envelopes for t-PTEN, n-PTEN, and 4p-PTEN overlaid with the tailless crystal structure containing the phosphatase domain (blue), C2 domain (magenta), and CBRIII loop (yellow). (**D**) Summary of $R_g$ and $D_{max}$ values for t-PTEN, n-PTEN, and 4p-PTEN.

The following figure supplements are available for figure 5:

**Figure supplement 1**. Molecular envelopes of t-PTEN, n-PTEN and 4p-PTEN obtained from SAXS analysis.

The radius of gyration ($R_g$) and maximum dimension ($D_{max}$) of the PTEN particles can be used as measurements of protein size. $R_g$ values calculated from the Guinier and p(r) plots are in agreement for each respective protein. Obtained from the p(r) plot, the $R_g$ for n-PTEN (31.91 +/− 0.054 Å) is larger than it is for 4p-PTEN (27.16 +/− 0.019 Å) and t-PTEN (27.82 +/− 0.083 Å). $D_{max}$ is also larger for n-PTEN (96.5 Å) than it is for 4p-PTEN (82.5 Å) and t-PTEN (82.5 Å) (*Figure 5D*).

Molecular envelopes of t-PTEN, n-PTEN and 4p-PTEN were generated from the scattering data using the ab initio modeling program DAMMIN (*Putnam et al., 2007*; *Franke and Svergun, 2009*; *Jacques and Trewhella, 2012*). Outputs from 10 DAMMIN runs, averaged for each protein using DAMAVER, are shown overlaid with the tailless crystal structure in *Figure 5C* (*Figure 5—figure Supplement 1*). The tailless crystal structure containing the phosphatase and C2 domains fits nicely into the two-lobed globular portion of each envelope. The molecular envelope for n-PTEN

reveals an elongated extension proximal to the C2 domain (*Figure 5C*). The molecular envelope for t-PTEN, which contains half of the 52 residue tail, shows a small extension, similar in length to that of 4p-PTEN, but perhaps not as wide. Interestingly, the 4p-PTEN phosphatase domain appears to undergo a modest change in shape relative to t-PTEN and n-PTEN (*Figure 5C* and *Figure 5—figure supplement 1*).

## Mutagenesis analysis of semisynthetic PTENs

Based on prior models as well as the proteolysis and SAXS experiments, we constructed several mutant semisynthetic PTENs to investigate possible PTEN residues that contribute to the closed conformation of 4p-PTEN (*Figure 6—figure supplement 1*). Semisynthetic phospho- and unphosphorylated PTEN containing four mutations K13A, R14A, R15A, and R161A ('N-mutant') was constructed to test the possibility that these basic residues in the N-terminus, co-localized in the crystal structure and implicated in phospholipid binding, might be critical in stabilizing the closed 4p-PTEN conformation. We also generated the penta-mutant unphosphorylated and phosphorylated PTENs containing K260A/K263A/K266A/K267A/K269A ('A5-mutant') and K260D/K263D/K266D/K267D/K269D ('D5-mutant') as neutral and charge inverted forms that probe the importance of the CBRIII loop in the C2 domain to conformational constriction. A fourth construct of semisynthetic PTEN that included deletions in the N-terminus (aa 1–6), D-loop (aa 286–309), and C-terminus (aa 395–403, and also contained T366A/S370A), ('X-mutant') was prepared to facilitate comparisons to the crystallized form of PTEN which possessed the same N-terminal and 'D-loop' deletions (*Lee et al., 1999*). These semisynthetic PTEN mutants were generated analogously to that of n-PTEN and 4p-PTEN. Anion exchange chromatography showed the same paradoxical behavior for the phosphorylated forms of the N-mutant, A5-mutant, and X-mutant PTENs with faster than expected elution of these semisynthetic proteins. Of note, elution of the unligated X-mutant PTEN on anion exchange chromatography sharpened to a single peak compared to the broader distribution of unligated wt PTEN (t-PTEN), presumably because mutation of Thr366/Ser370 abolishes the phosphorylation events that lead to heterogeneity (*Figure 6—figure supplement 2*). Moreover, N-mutant, A5-mutant, and X-mutant 4p-PTEN proteins showed nearly identical rates of dephosphorylation by alkaline phosphatase relative to wt 4p-PTEN indicating that they are each in the same ~25:1 equilibrium favoring the closed conformation (*Figure 6A* and *Figure 6—figure supplement 3*).

In contrast, the D5 mutant 4p-PTEN showed a distinctive biochemical behavior. With respect to anion exchange chromatography, D5 mutant 4p-PTEN eluted broadly with much of the protein eluting at higher NaCl concentrations than the unphosphorylated D5 mutant (*Figure 6—figure supplement 4*). Later elution is the expected profile for a typical phosphorylated protein relative to an unphosphorylated protein compared with the paradoxical pattern for n- and 4p- semisynthetic PTENs. In addition, D5 mutant 4p-PTEN was dephosphorylated about eightfold faster than the native wt 4p-PTEN and only threefold slower than denatured wt 4p-PTEN, implying that the tail phosphate cluster is dramatically more exposed in the D5 mutant (*Figure 6A* and *Figure 6—figure Supplement 3*). We considered the possibility that Asp substitutions of the Lys residues in the D5 mutant led to overall protein destabilization and denaturation. However, D5- mutant 4p-PTEN readily processed soluble $PIP_3$ substrate, ~threefold faster than wt and A5-mutant 4p-PTENs (*Figure 6B*). Moreover, the D5-mutant 4p-PTEN phosphatase activity showed saturation with soluble $PIP_3$ (*Figure 6C*), with catalytic parameters approaching those of n-PTEN, consistent with a more open conformation than that of wt 4p-PTEN. Taken together, these results make clear that the penta-Asp substitutions in the CBRIII loop are non-denaturing but can significantly disrupt the closed conformation of 4p-PTEN, presumably by electrostatic repulsion of the anionic tail. As reported previously (*Campbell et al., 2003*), the activity of the N-mutant which contains mutations of K13, R14, and R15 to alanine, is reduced compared to wt (*Figure 6B* and *Figure 6—figure supplement 5*).

## Intermolecular interactions between t-PTEN and C-tail phosphopeptide

We investigated the possibility that the synthetic C-tail tetraphosphorylated peptide (aa 379–403, 4p-25mer) used in ligation could modulate the activity of truncated (t-PTEN, aa 1–378) in an intermolecular fashion using the soluble $PIP_3$ substrate phosphatase assay. In this assay, t-PTEN showed similar behavior to n-PTEN, with saturable kinetics as a function of soluble $PIP_3$ (compare *Figures 7B and 2A*).

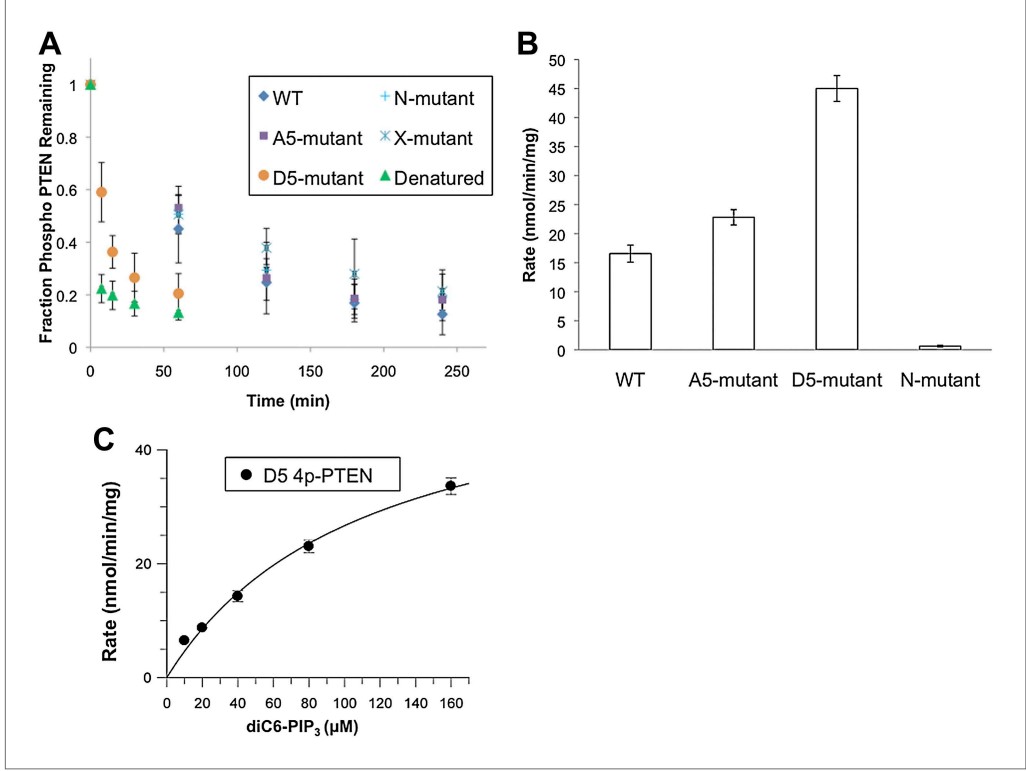

**Figure 6**. Phosphatase sensitivity and activity of 4p-PTEN and its mutants. (**A**) The rate of dephosphorylation of 4p-PTEN and its mutants was measured by quantification of bands from western blot analysis after treatment of the PTEN protein with 1 μM alkaline phosphatase. Data points are shown as the mean ± the SEM of three experiments. (**B**) PTEN activity was measured against 160 μM diC6-PIP$_3$ substrate. (**C**) $K_m$ curve of D5 4p-PTEN mutant ($k_{cat} = 3.0 \pm 0.3$ min$^{-1}$, $K_m = 112 \pm 22$ μM, $k_{cat}/K_m = 0.027 \pm 0.003$ min$^{-1}$μM$^{-1}$). Data points are shown as the mean ± the SEM of three experiments performed in duplicate.

The following figure supplements are available for figure 6:

**Figure supplement 1**. Schematic view of semisynthetic PTEN mutants.

**Figure supplement 2**. Anion exchange elution pattern of the PTEN X-mutant.

**Figure supplement 3**. Alkaline phosphatase sensitivity of 4p-PTEN and mutant forms.

**Figure supplement 4**. Anion exchange chromatography elution profiles of phosphorylated and unphosphorylated D5 PTEN.

**Figure supplement 5**. PTEN activity to diC6 PIP3.

Of note, the C-tail phosphopeptide 4p-25mer, but not the unphosphorylated peptide n-25mer, was a potent inhibitor of t-PTEN soluble PIP$_3$ phosphatase activity with an IC$_{50}$ ~ 1 μM (**Figure 7A**). In the presence of 1 μM 4p-25mer, the t-PTEN phosphatase activity showed an increase in $K_m$ for soluble PIP$_3$ (**Figure 7B**), mimicking the behavior of 4p-PTEN. Strikingly, the D5 mutant form of t-PTEN was resistant to the inhibition by 4p-25mer at concentrations up to 10 μM of the peptide (**Figure 7C**). Analogous to its effects on t-PTEN phosphatase activity, 4p-25mer but not n-25mer inhibited t-PTEN binding to PIP$_2$ vesicles, although less potently presumably because of competition for binding surfaces on the body of PTEN between the vesicle lipids and the 4p-25mer tail peptide (**Figure 7D**). Taken together, these experiments reveal that the intermolecular effects of the C-tail phosphopeptide on the PTEN body resemble those proposed for the intramolecular conformational change in 4p-PTEN.

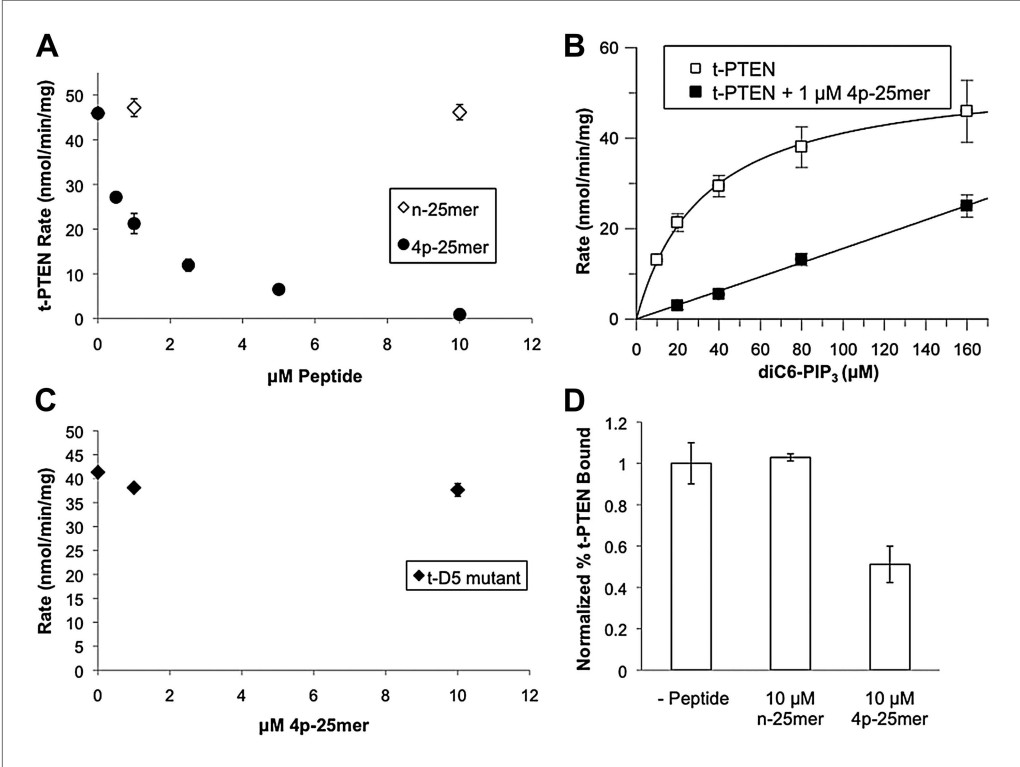

**Figure 7**. In trans peptide inhibition and binding of PTEN. (**A**) Tail peptide inhibition of t-PTEN with either n-25mer or 4p-25mer. (**B**) $K_m$ curve of t-PTEN in the absence ($k_{cat}$ = 2.9 ± 0.1 min⁻¹; $K_m$ = 33 ± 2.1 μM, $k_{cat}/K_m$ = 0.088 ± 0.004 min⁻¹μM⁻¹) or presence ($k_{cat}/K_m$ = 0.006 ± 0.0003 min⁻¹μM⁻¹) of 1 μM 4p-25mer phosphopeptide. (**C**) Reduced inhibition of D5 t-PTEN mutant in the presence of 4p-25mer peptide. (**D**) Vesicle sedimentation of t-PTEN in absence and presence of 10 μM tail peptides. Data points are shown as the mean ± the SEM of three experiments performed in duplicate.

## Discussion

The evidence developed here points to a straightforward model for how phosphorylation regulates PTEN structure and function. Upon phosphorylation on the 380–385 Ser/Thr cluster, the PTEN C-terminal modified tail clamps down intramolecularly on the C2 domain in the vicinity of the CBRIII loop, preventing PTEN from binding the plasma membrane and reducing its catalytic action toward $PIP_3$ (*Figure 8*). The combined structural data including ion exchange chromatographic behavior, trypsin protease susceptibility, alkaline phosphatase sensitivity, and SAXS analysis point to a more compact 4p-PTEN relative to n-PTEN in which the phosphates on the tail are concealed.

While it is formally possible that the phospho-tail of PTEN could cause indirect effects on the C2 domain by binding elsewhere on the PTEN body, the accumulated evidence argues for a direct phospho-tail–C2 interaction. The SAXS results suggest that phospho-tail is in close proximity to the C2 domain. Replacement of the PTEN Lys cluster of the CBRIII loop with Asp residues (5D mutant) but not Ala residues (5A mutant) in 4p-PTEN renders the phospho-tail more susceptible to alkaline phosphatase hydrolysis, presumably because the anionic phosphate tail clashes with the Asp carboxylate negative charges. The intermolecular effects of the 4p-25mer C-tail on the t-PTEN body and their reduced sensitivity conferred by CBRIII loop mutation further corroborate the phospho-tail–C2 interaction. Prior studies suggest that the five clustered Lys residues of the CBRIII loop are found on the membrane binding surface of the C2 domain, and their mutation decreases PTEN binding to lipid vesicles in vitro and plasma membranes in cell transfection experiments (*Lee et al., 1999*; *McConnachie et al., 2003*).

It is within the realm of possibility that 4p-PTEN can bind to membrane in its closed conformation, albeit with significantly reduced affinity. Our best evidence against the possibility that conformationally closed 4p-PTEN binds membrane is that the interfacial $K_m$s ($iK_m$s) for 4p-PTEN, n-PTEN, and t-PTEN are all the same, within error, in dephosphorylating vesicle-bound $PIP_3$ (*Figure 2*). Since the $K_m$s for

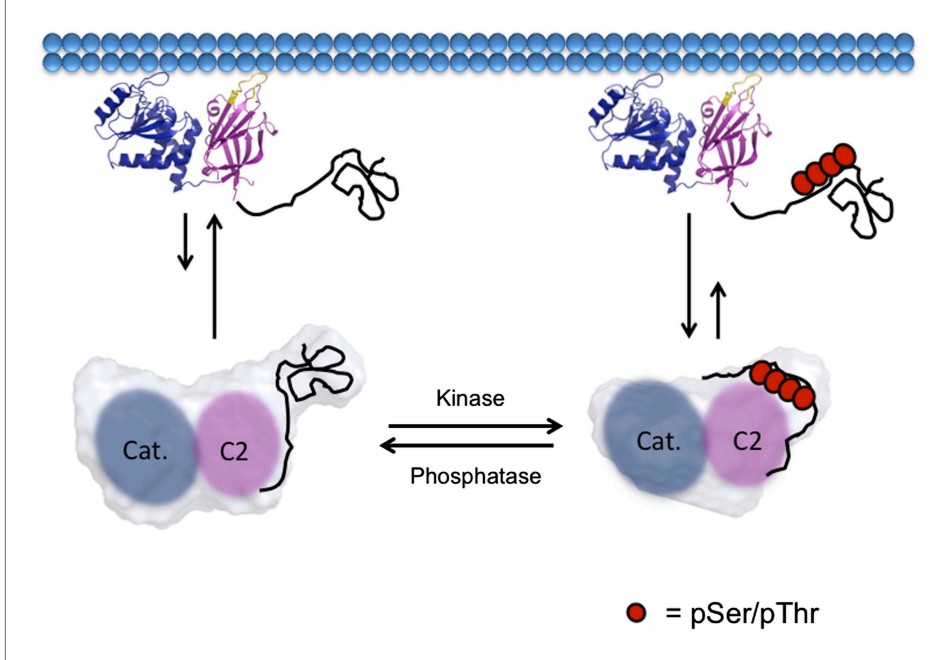

**Figure 8**. Model of PTEN regulation by phosphorylation. Upon phosphorylation, PTEN adopts a more compact conformation with the phosphorylated tail condensing around the CBRIII loop and membrane binding surface of the C2 domain, preventing it from binding to the plasma membrane. When dephosphorylated, the tail of PTEN is no longer bound tightly to the C2 domain, allowing for the open PTEN protein to bind efficiently to the plasma membrane. Both phosphorylated and unphosphorylated PTEN are in the same open conformation when bound to the plasma membrane.

soluble $PIP_3$ dephosphorylation are different for 4p-PTEN and n-PTEN (*Figure 2*), it would be surprising that the $iK_m$s would be the same if the conformation of 4p-PTEN were closed when bound to vesicle. It is plausible, however, that some closed 4p-PTEN with reduced membrane affinity is interacting with vesicle while in a catalytically impaired state, which would contribute only in a minor way to the $iK_m$. Nevertheless, we also know how important the CBRIII loop is for membrane binding which, as discussed above, is likely abutting the phospho-tail in the closed conformation. Thus, the information available strongly supports the model in *Figure 8*.

We observed here that C-terminal phosphorylation of PTEN significantly reduces the enzyme's lipid bilayer affinity and also its catalytic efficiency with soluble $PIP_3$ substrate and at low (<1 mol%) $PIP_3$ surface concentrations in vesicles. Since physiologic membrane surface concentrations of $PIP_3$ are in the range of 0.001 mol% or less, C-terminal phosphorylation of PTEN should confer an important reduction in enzymatic activity in vivo (*Rahdar et al., 2009*). It is interesting that the soluble $PIP_3$ $K_m$ for 4p-PTEN vs n-PTEN is elevated whereas the interfacial $K_m$ ($iK_m$) with vesicles containing $PIP_3$ is the same for both enzyme forms. Both the trypsin protease pattern and SAXS analysis of 4p-PTEN vs n-PTEN reveal apparent structural/dynamic changes within the catalytic domain upon phosphorylation that may account for the altered phosphatase properties. Such conformational changes in the catalytic domain upon phosphorylation may come from direct interactions with the tail, though they could also be transmitted allosterically through tail-C2 binding since the C2 domain makes intimate contacts with the catalytic domain. It should be noted that the CBRIII loop, while critical for vesicle interaction, is dispensable for soluble $PIP_3$ substrate processing, as shown here and previously (*McConnachie et al., 2003*). Thus, there are at least some differences between interactions of PTEN with soluble $PIP_3$ and membrane-embedded $PIP_3$.

Results from previous experiments involving co-immunoprecipitation of the co-transfected PTEN tail and body as separate pieces hinted at the potential of a conformational change induced by phosphorylation (*Rahdar et al., 2009*). However, in contrast to the findings here, the co-immunoprecipitation of the PTEN tail and body was disrupted by the altered residues in the N-mutant and A5-mutant

PTENs. Since the N-mutant and A5-mutant 4p-PTENs showed the same apparent closed equilibrium constant as wt 4p-PTEN, this suggests that there may be differences in this trans interaction identified in mammalian cell extracts vs the closed conformation of the intact 4p-PTEN molecule analyzed in the current study. Indeed, we found higher than expected apparent affinity between the 4p-25mer and the t-PTEN body ($IC_{50}$ 1 μM) and more complete inhibition (>95%) of the lipid phosphatase activity of t-PTEN by the 4p-25mer with soluble $PIP_3$ substrate than would be predicted based on results with 4p-PTEN (4p-PTEN:n-PTEN activity was ~6:1). The ~25:1 equilibrium favoring the closed conformation of 4p-PTEN based on alkaline phosphatase susceptibility seems relatively small compared with an apparent $K_d$ of ~1 μM for the 4p-25mer-t-PTEN intermolecular interaction. Such results may suggest structural differences in the intermolecular complex vs the intramolecular conformational change. While there are several plausible explanations for this, one interesting possibility is that there may be energetic strain associated with achieving conformational closure in the intramolecular case of 4p-PTEN not associated with a phospho-tail–t-PTEN intermolecular interaction.

Long-range intramolecular protein conformational switching induced by phosphorylation has been observed in several well-established cases including CrkL, Src, and SHP-1/2 (*Lu et al., 2001*; *Rosen et al., 1995*; *Sicheri et al., 1997*; *Xu et al., 1997*; *Zhang et al., 2003*). Each of these examples involves an SH2 domain interacting with a phosphotyrosine. We propose a Src-like model for PTEN, in which a cluster of pSer/pThr drives an apparently long distance intramolecular binding interaction to deactivate the enzyme. How 4p-PTEN may be reactivated in vivo remains an important question. There are presumably cellular phosphatase enzymes that can dephosphorylate phospho-PTEN or ligands which can bind allosterically to phospho-PTEN that may promote conformational opening. In contrast to prior suggestions based on trans experiments, we see no evidence of the autodephosphorylation of 4p-PTEN (*Zhang et al., 2012*). As we are reliant on western blots with an antibody that may recognize states of depleted phosphorylation of the tail, partial autodephosphorylation cannot be completely ruled out by our experiments. However, the stability of the closed conformation of 4p-PTEN over the dozens of hours of expressed protein ligation points to resistance of the tail phosphates to PTEN enzymatic removal. Because of the structural concealment of the 380–5 phosphoSer/Thr residues in the closed 4p-PTEN conformation, such autodephosphorylation may be especially disfavored.

There are several biomedical implications of the findings here. Altered protein–protein interactions or efficiency of ubiquitylation may be influenced by phosphorylation-mediated conformational closure of phospho-PTEN. While PTEN is a tumor suppressor and can be mutated in cancer, it is often wild type but expressed at low levels. Direct stimulation of cellular phospho-PTEN by pharmacologic agents could prove to be effective as an anti-cancer therapy. A related approach has been explored to activate the tumor suppressor p53 (*Foster et al., 1999*) and the apoptotic protein procaspase (*Gray et al., 2010*). It may be possible to find small molecules that bind specifically to the phospho-tail and prevent its intramolecular engagement with the C2 domain or bind somewhere on the PTEN body and stabilize the open PTEN conformation allosterically. The soluble $PIP_3$ substrate dephosphorylation assay with 4p-PTEN should provide a means for screening for such activators. Alternatively, inhibitors of CK2 and/or GSK3β protein kinase, the enzymes responsible for PTEN C-tail phosphorylation, may be effective in targeting $PIP_3$/Akt-driven tumors.

This investigation also highlights the use of intein-mediated protein thioester formation in an insect cell expression system for investigation of post-translational modifications. Multi-site phosphorylation has been elegantly studied in the TGFβ signaling axis using protein semisynthesis previously (*Huse et al., 2001*; *Wu et al., 2001*), but these experiments did not require insect cell expression of an intein-fusion protein. Our efforts here suggest that baculovirus expression systems are an attractive option, rather than a last resort, for expressed protein ligation with challenging eukaryotic proteins.

## Materials and methods

### Reagents

All lipids were from Avanti Polar Lipids (Alabaster, AL). Antibodies were from Santa Cruz Biotechnology (Dallas, TX) (SC-6818) and Novus Biological (Littleton, CO) (NBP1-4136 and NBP1-44,412). MESNA was from Sigma (St. Louis, MO). All Fmoc-amino acids were from EMD (Billerica, MA).

### Peptide synthesis

All peptides were synthesized on a PS3 peptide synthesizer from Protein Technologies (Tuscon, AZ) or by hand using Fmoc based standard solid phase peptide synthesis.

## Generation and purification of semisynthetic PTEN

PTEN C-terminally truncated at residue 378 was first subcloned into the pTXB1 vector from NEB which contains the GyrA intein from the organism *Mycobacterium xenopi*. Tyr 379 was mutated to a Cys to facilitate the intein mediated cleavage reaction. The PTEN-intein-cbd DNA sequence was then subcloned into the pFastBac1 baculovirus entry vector and the subsequent baculovirus was generated. The PTEN-intein-cbd fusion protein was expressed in HighFive insect cells.

The fusion protein in 40 ml of cell lysate from 1 l of cell culture was first passed over a 10 ml bed of fibrous cellulose (Whatman) to remove viral chitinase, then bound to a 6 ml bed of chitin beads from NEB in a gravity flow chromatography column from Bio-Rad (2.5 cm diameter). The chitin bead column with fusion protein bound was then washed with 250 ml of washing buffer (50 mM HEPES pH 7.6, 250 mM NaCl, 0.1% Triton X-100). C-terminally truncated PTEN (t-PTEN) was generated by DTT cleavage of the fusion protein, producing t-PTEN at yields of ~8–10 mg per liter of cell culture. Full length semisynthetic PTEN was then generated on the chitin column by adding 400 mM MESNA and 2 mM of C-terminal peptide buffered with 3 ml of 50 mM HEPES (pH 7.2), 150 mM NaCl. Based on the post ligation yield of PTEN protein it is estimated that there is a maximum PTEN-thioester concentration of ~80 µM in the ligation reaction. The ligation reactions were carried out for 48–72 hr at room temperature and monitored by SDS PAGE. Upon completion of the ligation reaction the ligation mixture was eluted from the chromatography column with 15 ml of dialysis buffer (50 mM Tris pH 8.0, 150 mM NaCl, 10 mM DTT) and subsequently dialyzed into 4 l of dialysis buffer for a period of 48 hr with multiple buffer exchanges in a dialysis cassette (Slidealyzer) with a 12K MWCO in order to remove excess unreacted peptide. Proteins were then concentrated following dialysis to >5 mg/ml (>100 µM). We estimate that there is <10 µM of residual unligated peptides at this stage. Due to large dilutions (>1000-fold) of the semisynthetic enzyme for enzymatic and other biochemical assays, small amounts of residual contaminating peptide remaining after dialysis would not be expected to interfere with any assays. Semisynthetic PTEN proteins produced in this way yields 8–10 mg of protein per liter of insect cell culture with the desired modifications on the C-terminus at purities of >90% based on Coomassie stained SDSPAGE. The semisynthetic protein was further purified for SAXS and soluble substrate assays by anion exchange chromatography (monoQ) using an AKTA FPLC from GE Healthcare. Proteins were purified with a gradient of 0–50% Buffer B over 250 ml at a flow rate of 1.0 ml/min (Buffer A: 50 mM Tris pH 8.0, 10 mM DTT; Buffer B: 50 mM Tris pH 8.0, 1.0 M NaCl, 10 mM DTT). After FPLC purification by anion exchange chromatography and concentration the final yield of semisynthetic PTEN protein was 2–3 mg per liter of cell culture with estimated purity >95% by coomassie stained SDS-PAGE. Size-exclusion chromatography was carried out with a Superdex 200 column in the following buffer: 50 mM Tris pH8.0, 150 mM NaCl, 10 mM DTT.

## Generation of radiolabed PIP$_3$

Radiolabeled PIP$_3$ was generated as previously described (*McConnache et al., 2003*). Briefly, PIP$_3$ labeled with $^{32}$P$_i$ at the three position of the inositol ring was generated by incubating PI3K (Echleon) and PIP$_2$:PS (1:1) vesicles in the presence of 250 mCi $^{32}$P-ATP, 1 mM ATP and 2.5 mM MgCl$_2$ in PI3K assay buffer (25 mM HEPES pH 7.6, 120 mM NaCl and 1 mM EGTA). After a Bligh-Dyer extraction of PIP$_3$, thin-layer chromatography (TLC) showed radiolabeled PIP$_3$ to be the major product (*Figure 2—figure supplement 3*). TLC solvent conditions: (CHCl$_3$:Acetone:MeOH;Acetic Acid:H$_2$0); (70:20:50:20:20).

## Vesicle based phosphatase assays

Lipid phosphatase assays were modified from those already described in the literature (*McConnachie et al., 2003*). $^{32}$P radiolabeled PIP$_3$ was incorporated into vesicles containing unlabeled PIP$_3$, phosphatidylcholine (PC), and/or phosphatidylserine (PS) and/or PIP$_2$ by sonication of dried lipids hydrated in the presence of PTEN assay buffer (50 mM Tris pH 8.0, 150 mM NaCl, 10 mM DTT, 1 mM EGTA). Lipids were sonicated in 100 µl volumes in glass test tubes at room temperature until the solution clarified. Vesicles made in this way are 30–50 nm in diameter. Vesicles were used in assays within 15 min of being made. In assay reactions, the ratio of the number of vesicles to the number of PTEN molecules was maintained at 4:1 or greater. Ovalbumin (0.05 mg/ml) was used to stabilize the PTEN protein in the assays. 25 µl reactions were initiated by the addition of vesicle substrate and incubated at 30°C for 3 min. The reaction was quenched with 3 M perchloric acid. Hydrolyzed $^{32}$P$_i$ was then separated from $^{32}$P-PIP$_3$ by a Bligh–Dyer extraction. The aqueous phase was then treated with 1% ammonium

molybdate and the resulting phosphate–molybdate complex was extracted with toluene:isobutanol (1:1). This organic phase was then counted using a Beckman scintillation counter.

## Interfacial kinetic analysis

Analysis of the kinetic parameters of the semisynthetic PTEN proteins were determined in accordance with the procedures pioneered by Dennis and coworkers and previously performed for recombinant PTEN produced in *E. coli* (*Deems et al., 1975*; *Hendrickson and Dennis, 1984*; *McConnachie et al., 2003*). With this type of analysis, the initial velocity of an interfacial enzyme follows the equation below:

$$V_0 = (V_{max} * X_s * [S_0]) / (iK_m * K_s + iK_m * [S_0] + X_s * [S_0])$$

Two types of experiments were performed, bulk dilution (BD) and surface dilution (SD). In BD experiments, the surface concentration of $PIP_3$ was held constant and bulk concentration of $PIP_3$ was varied by varying the concentration of $PIP_3$ and the carrier lipid PC proportionately. In SD experiments, the bulk concentration of $PIP_3$ was held constant and the surface concentration was varied by varying the amount of PC. In both types of experiments rectangular hyperbolas were obtained with apparent $V_{max}$ and apparent $K_m$ values. Apparent $V_{max}$ and apparent $K_m$ values were then fit to the equations below to determine the kinetic variables for each PTEN protein.

$$ik_m = (V_{max\ SD}/V_{max\ BD} - 1)\,X_s$$
$$K_s = K_{m\ BD}\,(X_s/iK_m + 1)$$
$$k_{cat} = V_{max\ SD} * [E_T]$$

## Vesicle pulldowns

Large Multilamillar Vesicles (LMVs) containing various amounts of PC, PS and/or $PIP_2$ were generated by vigorously vortexing dried lipids that were hydrated in the presence of PTEN buffer for 5 min in 1 ml volumes. The LMVs were then incubated with different forms of PTEN protein for 30 min at 25ºC. The vesicles and bound protein in 50 µl volumes were then pelleted at 180,000*g* using a Beckman ultracentrifuge for 2 hr. The supernatant was removed from the vesicle pellet, the pellet washed with buffer, then boiled in 10% SDS loading dye and run on SDS-PAGE. The amount of PTEN protein that bound to the LMVs was then visualized by western blot using an anti-PTEN antibody from Santa Cruz Biotechnologies (SC-6818). The amount bound was quantified using Carestream Media image quantification software. For tail peptide competition assays the amount of tail peptide used was 10 µM.

## Trypsin digests

2 µg of semisynthetic PTEN in 20 µl reactions volumes was digested with varying amounts of trypsin (Promega, V511A) for 10 min at 37°C in PTEN assay buffer. Reactions then were quenched with SDS loading dye and run on SDS-PAGE. The digestion fragments were visualized by Colloidal Blue stain from Invitrogen (LC6025) or by western blot (antibodies SC-6818 or NBP1-44,412).

## Sequencing of trypsin digest fragments

Trypsin digestion products were run on a 10% SDS-PAGE gel and transferred to a PVDF membrane. The membrane was then stained with Coomassie stain. The bands of interest were cut out of the membrane and analyzed by N-terminal Edman degradation sequencing at the JHMI Synthesis and Sequencing Facility.

## Phosphatase sensitivity assay

50 ng of semisynthetic phosphorylated PTEN and its mutants were dephosphorylated in the presence of 1 µM alkaline phosphatase from NEB (CIP) for varying periods of time in phosphatase assay buffer (50 mM Tris pH 8.0, 20 mM NaCl, 25 µM $MgCl_2$ and 10 mM DTT) at room temperature in 20 µl. Dephosphorylation of PTEN was monitored by western blot with an antibody to the phospho-tail cluster (NBP1-4136). The fraction of phospho-PTEN remaining was determined using Carestream Media image quantification software.

## Soluble substrate activity assay

PTEN activity to a water soluble substrate (diC6 $PIP_3$) was determined by measuring the release of inorganic phosphate with a malachite green (*Van Veldhoven and Mannaert, 1987*) detection kit from R and D Biosystems. 25 µl reactions were allowed to proceed for 5–10 minutes at 30°C in assay buffer

(50 mM Tris pH 8.0, 10 mM BME) before being quenched by malachite green reagent. Amounts of PTEN used per data point ranged from 0.5 to 20 μg. Reactions were shown to be linear with respect to time and enzyme concentration in the ranges used. For in trans peptide inhibition assays, the amount of tail peptide used (quantified by amino acid analysis) is indicated in the figure legend. It is unlikely that any peptide ligated to t-PTEN given the low concentrations of peptide used, short reaction times and the low reactivity of the DTT-thioester toward native chemical ligation. In fact, with the vesicle pulldowns that used 10 μM phosphopeptide and t-PTEN, there is no evidence of ligation observed by western blot.

### Small-angle X-ray scattering (SAXS)

SAXS experiments were performed at Brookhaven National Laboratories at the National Synchrotron Light Source (NSLS), beamline X9 using a MarCCD detector, located 3.4 m from the sample. Data for each protein sample was collected in triplicate. All samples were in PTEN assay buffer. 20 μl of each sample was continuously passed through a capillary tube exposed to a 400 × 200 μm X-ray beam and data recorded for 30s. Normalization for beamline intensity, buffer subtraction and merging of data were carried out using proc.py software developed by the beamline staff (*Allaire and Yang, 2011*).

SAXS data analysis was carried out using software from the ATSAS program suite. The radius of gyration ($R_g$) was calculated using a Guinier approximation with the program PRIMUS (*Konarev et al., 2003*). The pair distribution function P(r) and the maximum particle dimension ($D_{max}$) were determined using GNOM (*Svergun et al., 1992*). Ten ab initio models were generated for each protein using DAMMIN, then averaged using DAMAVER (*Volkov and Svergun, 2003*; *Putnam et al., 2007*; *Franke and Svergun, 2009*). The resulting molecular envelopes were fit with the tailless crystal structure in PyMOL. Figures of the models were made using PyMOL.

## Acknowledgements

We thank T Woolf, MK Tarrant, and M Allaire for assistance and helpful advice. We are grateful to the NIH for support.

## Additional information

### Competing interests

PC: Reviewing editor, *eLife*. The other authors declare that no competing interests exist.

### Funding

| Funder | Grant reference number | Author |
| --- | --- | --- |
| National Institutes of Health | | Philip Cole, Peter Devreotes |
| National Institutes of Health | 4R37CA043460 | Sandra B Gabelli, L Mario Amzel |

The funders had no role in study design, data collection and interpretation, or the decision to submit the work for publication.

### Author contributions

DB, MR, SBG, Conception and design, Acquisition of data, Analysis and interpretation of data, Drafting or revising the article; BT-S, Acquisition of data, Analysis and interpretation of data, Drafting or revising the article; SCS, Acquisition of data, Drafting or revising the article, Contributed unpublished essential data or reagents; DR, LMA, PD, PC, Conception and design, Analysis and interpretation of data, Drafting or revising the article

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
