## [Decision Letter]

[Editors' note: this article was originally rejected after peer review but a revision was invited after the concerns of the reviewers were addressed with new experimental data.]

Thank you for choosing to send your work entitled “Phosphorylation-mediated PTEN Conformational Closure and Deactivation Revealed with Protein Semisynthesis” for consideration at *eLife*. Your article has now been peer reviewed and we regret to inform you that the current submission will not be considered further for publication at this time. Your submission has been evaluated by 3 reviewers, one of whom is a member of our Board of Reviewing Editors, and a Senior editor, and the decision was reached after discussions between the reviewers.

As you will see below, all three reviewers were enthusiastic about the application of protein semisynthesis towards studies on the role of phosphorylation in the regulation of PTEN structure and localization. However, specific concerns were raised in two of the reviews that make the manuscript unsuitable for publication in *eLife* at the current time. We believe that the reviewers’ comments and the post-review discussion with Reviewer 3 provide useful information as to how the manuscript could be revised to address the concerns that have been raised and we would invite you to consider whether it would be possible for you to revise the manuscript to address these comments in a reasonable period of time.

Reviewer #1:

This manuscript presents studies that seek to gain insight into the mechanistic basis of phosphorylation-mediated PTEN regulation. The approach centers on the development and biochemical and biophysical analysis of a semisynthetic PTEN variant that is uniformly and uniquely modified with four phosphorylated serine and threonine residues in the C-terminal tail.

I find this to be a significant contribution for the following reasons:

1) The research specifically addresses the physical basis for the effects of PTEN phosphorylation with respect to whether the modifications directly alter the membrane association properties of the protein or whether they cause a change in inter- or intramolecular interactions, which result in changes membrane association.

2) The uniquely phosphorylated proteins provide information that is not readily accessible via mutational analysis with encoded amino acids or via enzyme-catalyzed phosphorylation since one cannot readily achieve uniquely phosphorylated materials.

3) Multiple lines of evidence are presented for a model (Figure 7) in which the tail-phosphorylated PTEN (4p-PTEN) more favorably adopts a compact structure, which may feature the negatively-charged tail in a buried state that reduces plasma-membrane binding. In contrast the non-phosphorylated n-PTEN is preferentially in a more open state making the membrane-associating face of PTEN more available for binding to membrane.

Concerns:

1) The authors should include the protein concentration in the ligation reaction – the yields are impressive so it is useful to know this detail.

2) It is noted in the methods that the PTEN-intein-cbd is produced in HighFive insect cells at a concentration of 10 mg/L culture – and then further noted that the final ligated semisynthetic proteins are produced in yields of ∼ 10 mg/L insect cell culture (at >90% purity). It is not clear how this is possible what with the loss of mass from intein-cbd cleavage and the subsequent manipulations related to the thioester exchange, ligation, and purification. This needs to be clarified as it speaks to the next issue that concerns the purity of the semisynthetic protein products. Additionally, there is no detail provided on how the semisynthetic material is purified after the ligation and how are the unligated protein and excess peptide removed?

3) It is noted that western blot analysis with the “anti-phospho-PTEN” Ab confirms that the final material is appropriately phosphorylated. Is it certain that this anti-P Ab only recognizes the fully tetra-phosphorylated C-terminus? One might imagine that constructs with reduced phosphorylation might also be recognized. This question also comes up in the context of studies on auto-dephosphorylation. Perhaps, some erosion of the total phosphorylation is acceptable – but the authors should definitely comment here.

4) The authors should state on the yield after anion exchange chromatography that affords extra pure material for SAXS. This would provide information on the purity of the material directly from the ligation reaction. This issue is particularly important when considering and comparing the catalytic parameters and vesicle binding of 4p-PTEN.

5) It would be useful to contrast the current work with the Muir study on the type I TGF-beta receptor soluble domain wherein a hyperphosphorylated N-terminal tail is attached to the protein via semisynthesis and Smad2 activation is studied. The authors do cite reviews in which theses studies are highlighted, but it would be appropriate to highlight these particular examples wherein hyperphosphorylated sequences are incorporated into proteins to study structure and function.

Reviewer #2:

This is a superb study of the biophysical properties of the protein phosphatase PTEN. I really could find no issues with the report.

Reviewer #3:

The protein ligation technique is very useful for studies on site specific phosphorylation, and the Cole lab has used this method cleverly to interrogate the function of PTEN phosphorylation. The methods used are appropriate and described thoroughly and the data collection is sound.

1) The main weakness is that while phosphorylation leads to protein compaction, reduced catalysis and reduced lipid affinity, the study lacks evidence that the inhibitory effects on activity and/or binding are necessarily caused by the conformational changes reflected by SAXS and proteolysis. Without this evidence, the autoinhibitory model shown in Figure 7 is speculative, which reduces the impact of the study.

2) Figure 2 suggests that 4p-PTEN might reduce k_cat_, as well as k_cat_/K_m_. More evidence is needed proving that the effect of phosphorylation on catalysis can be explained only by reduced phospholipid binding. If possible, k_cat_ should be measured with less error (e.g., using alternative substrates), in order to eliminate the possibility that phosphorylation affects turnover rate as well as binding constants, which would indicate mechanisms beyond simple steric interference with the active site.

Additional comments from Reviewer 3 based on the discussion between the reviewers:

Discussion on comment 1: With respect to the presented “autoinhibitory model” it would be possible for the authors to indicate that the model that is put forward in the paper is one of many possibilities, and perhaps present hypotheses for alternate models that might be consistent with their observations.

Discussion on comment 2: The additional work that would be needed to demonstrate autoinhibition would require (1) showing similar kinetic parameters between t-PTEN and n-PTEN, and (2) showing that the phosphorylated C-terminal peptide inhibits t-PTEN in trans down to an activity comparable to 4p-PTEN. In order to argue that autoinhibition works by interfering with lipid binding, evidence would be needed showing that (3) the phospho-C-terminal peptide interacts with the lipid-binding interface, and (4) that conservative mutations at the lipid-binding interface interfere with peptide binding and relieve autoinhibition of 4p-PTEN. Other strategies might also be suitable, but these are among the experiments that I believe were used to demonstrate autoinhibition in Src kinases.

The question about k_cat_ could readily be addressed using p-nitrophenylphosphate or perhaps another short chain acyl PI3,4,5P3 (e.g., C2, C3, C4) with greater solubility in aqueous solution.

---

## [Author Response]

Although we were heartened by the positive statements, we appreciate the seriousness of the critiques that necessitated our revisions to better explain and bolster the evidence for the linkage of structural and functional change in our phospho-PTEN model of old Figure 7 (now Figure 8 in the revised manuscript). We understood that the major missing piece from the reviewers’ perspective relates to whether the C-tail phosphopeptide of PTEN can act in trans to impact the lipid phosphatase activity of the body of PTEN (t-PTEN). We have now performed the key experiments that address this concern. In brief, the results are fully consistent with the (new) Figure 8 model. We have found that t-PTEN shows similar enzymatic activity to n-PTEN with the soluble PIP_3_ substrate. C-tail phosphopeptide potently inhibited t-PTEN lipid phosphatase activity with soluble PIP_3_ and t-PTEN binding activity to PIP_2_ containing-lipid vesicles, whereas the unphosphorylated C-tail had no effect in the same concentration range. Moreover, the C2 domain mutant form of t-PTEN (D5 t-PTEN, which contains mutations on the membrane binding CBRIII loop) was resistant to the catalytic inhibition effects of C-tail phosphopeptide, consistent with the model that a principal functional interaction between the body and phospho-tail involves the lipid binding interface of the C2 domain. In addition, we have also performed a more complete analysis of the substrate concentration dependence of the 5D mutant of 4p-PTEN lipid phosphatase activity, and these results fortify the interpretation of the regulatory implications of this mutant. These data and a related discussion are now added to the revised manuscript (new Figure 6 and new Figure 7). We think the inclusion of these results significantly strengthens the manuscript and establishes a modular structural interaction for phospho-PTEN. Below we delineate our point-by-point responses to the reviewers’ comments and describe the accompanying changes in the revised manuscript.

Reviewer #1:

*1) The authors should include the protein concentration in the ligation reaction – the yields are impressive so it is useful to know this detail*.

The expressed protein ligation reaction was carried out in a one-pot procedure in which the cell extracts were loaded on to a chitin-resin affinity column, and washed to remove impurities. This resin-bound PTEN-intein-CBD fusion protein was ligated with 2 mM N-Cys peptide and the solution phase protein contents were separated from the chitin-resin. We have added to the revised methods the approximate ratio of resin volume used to cell lysate volume as well as the reaction vessel that was employed. We never isolate the thioester species in this procedure, to minimize manipulations and to reduce losses from thioester hydrolysis, so we do not know the precise PTEN thioester fragment protein concentration present at the moment of ligation. We can, however, estimate that there is a maximum of 80 μM of protein thioester in the reaction mixture based on the recovered product.

*2) It is noted in the methods that the PTEN-intein-cbd is produced in HighFive insect cells at a concentration of 10 mg/L culture – and then further noted that the final ligated semisynthetic proteins are produced in yields of ∼ 10 mg/L insect cell culture (at >90% purity). It is not clear how this is possible what with the loss of mass from intein-cbd cleavage and the subsequent manipulations related to the thioester exchange, ligation, and purification. This needs to be clarified as it speaks to the next issue that concerns the purity of the semisynthetic protein products. Additionally, there is no detail provided on how the semisynthetic material is purified after the ligation and how are the unligated protein and excess peptide removed*?

We regret that the discussion of yield was presented in a confusing fashion. As noted above, we never isolate/characterize the PTEN-intein-CBD fusion protein (CBD-chitin binding domain) or quantify its yield. The 10 mg/L of PTEN protein from HighFive cell culture is based on DTT cleavage from this resin to liberate the thioester protein, but is somewhat variable from run to run so we have changed the revised text to reflect that the yield is in the range of 8–10 mg/L. The protein purity of semisynthetic PTEN is routinely greater than 90% based on SDSPAGE after dialysis and concentration by ultrafiltration. Excess synthetic peptide (MW 3 kDa) was removed by extensive dialysis (1:400 vol:vol) over 24–48 hours with at least 3 buffer exchanges using 12 kDa MW cutoff pre- moistened dialysis membrane (Slidealyzer), as well as during concentration with a 10 kDa MW cutoff concentrator to >5 mg/ml (>100 μM) semisynthetic PTEN.

We estimate based on these procedures and appearance of SDSPAGE that less than 10 μM peptide remained in stored semisynthetic PTEN protein samples. Since vesicle binding and interfacial enzymatic assays were routinely carried out with >1000-fold dilution of the stock concentration of semisynthetic PTEN, possible impact of contaminating trace peptide is assumed to be negligible. As discussed below, the IC50 of phosphopeptide with t-PTEN is ∼1 μM and the contaminating phosphopeptide final concentration in 4p-PTEN- containing assays is estimated to be less than 10 nM. For both SAXS and soluble PIP_3_ phosphatase assays where concentrations of semisynthetic PTENs were greater than 500 nM, anion exchange chromatography was also used to further purify the protein and eliminate trace unligated peptide. Further details are added to the revised manuscript to include this information.

*3) It is noted that western blot analysis with the “anti-phospho-PTEN” Ab confirms that the final material is appropriately phosphorylated. Is it certain that this anti-P Ab only recognizes the fully tetra-phosphorylated C-terminus? One might imagine that constructs with reduced phosphorylation might also be recognized. This question also comes up in the context of studies on auto-dephosphorylation. Perhaps, some erosion of the total phosphorylation is acceptable – but the authors should definitely comment here*.

These are astute points. As Reviewer 1 suspects, the anti-phospho-PTEN Ab is unlikely to be specific for the tetraphosphate form, although this has not been systematically investigated. The antibody was raised against phospho S380/T382/T383. It certainly appears to recognize tetraphosphorylated PTEN. It is unknown if it will recognize phosphorylation states less than the three it was generated against. We have revised the manuscript to include this point, and also now state that we cannot rule out that some auto-dephosphorylation of the tail is formally possible because of the limitations of the western blot method.

However, it should be emphasized that the conformation of 4p-PTEN appears stable over dozens of hours (during ligation at room temperature) so that partial auto-dephosphorylation, if it is occurring, is non-disruptive. Furthermore, as noted in the manuscript, mass spectra of the intact 4p-PTEN and n-PTEN are consistent with a full complement of four phosphates in the former, based on the mass increase of 320 Da in 4p-PTEN relative to n-PTEN, although admittedly this is only semi-quantitative. However, in our experience phosphorylation of a peptide or protein tends to suppress MALDI ionization so that reduced phosphorylation state impurities would typically give larger mass spec signals.

*4) The authors should state on the yield after anion exchange chromatography that affords extra pure material for SAXS? This would provide information on the purity of the material directly from the ligation reaction. This issue is particularly important when considering and comparing the catalytic parameters and vesicle binding of 4p-PTEN*.

We estimate that the purity of the semisynthetic PTEN protein after ligation is >90% and after anion exchange chromatography is >95%. The yield after anion exchange chromatography from 1 L of cell culture is estimated to be 2–3 mg/mL. We have made this clarification in the revised manuscript.

*5) It would be useful to contrast the current work with the Muir study on the type I TGF-beta receptor soluble domain wherein a hyperphosphorylated N-terminal tail is attached to the protein via semisynthesis and Smad2 activation is studied. The authors do cite reviews in which theses studies are highlighted, but it would be appropriate to highlight these particular examples wherein hyperphosphorylated sequences are incorporated into proteins to study structure and function*.

We agree that the prior elegant work from Tom Muir’s group on TGF-beta kinase and Smad2 merits explicit discussion and citation in our manuscript, and we have added this to the Discussion in the revised manuscript. As Reviewer 1 notes, both the enzyme (TGF-beta kinase) and substrate (Smad2) were prepared in multi-phosphorylated forms using semisynthesis methods. One key technical difference between the TGF-beta/Smad2 studies and the current PTEN work is that insect cell expression with a fused intein was not required in the Muir experiments.

Reviewer #3:

*1) The main weakness is that while phosphorylation leads to protein compaction, reduced catalysis and reduced lipid affinity, the study lacks evidence that the inhibitory effects on activity and/or binding are necessarily caused by the conformational changes reflected by SAXS and proteolysis. Without this evidence, the autoinhibitory model shown in Figure 7 is speculative, which reduces the impact of the study*.

To summarize how we arrived at the (new) Figure 8 model in the manuscript, we delineate the key points here. 1) By comparing n-PTEN and 4p‐PTEN, we found that tail phosphorylation of PTEN reduces catalytic efficiency with a soluble PIP_3_ substrate, at least in part by elevating the K_m_; 2) n-PTEN, 4p‐PTEN (and t-PTEN) show similar interfacial K_m_ values (i*K*_m_s) with vesicle embedded PIP_3_ substrate (suggesting similar conformations among these three forms of PTEN when bound to membrane) but 4p-PTEN shows relatively diminished vesicle affinity as judged by enzyme kinetic analysis as well as sedimentation experiments; 3) Relative to n-PTEN, 4p-PTEN shows a paradoxical weakened affinity for anionic chromatography resin, resistance to proteolysis, and more compact SAXS structure; 4) 4p-PTEN C-terminal phosphates are concealed in the native state based on susceptibility to alkaline phosphatase; 5) Relative to wt 4p-PTEN, the D5 (five clustered Lys residues of the membrane binding CBRIII loop replaced by Asp residues) 4p-PTEN mutant is conformationally partially open based on its enhanced susceptibility to alkaline phosphatase and more normal mobility in anion exchange chromatography; 6) The lipid phosphatase activity of the D5 4p-PTEN mutant with soluble PIP_3_ substrate is elevated relative to wt 4p-PTEN and is similar to the activity of wt n-PTEN, which has an open conformation (by SAXS, proteolysis, anion exchange chromatography); 7) The SAXS molecular envelopes calculated for 4p-PTEN, n-PTEN, and t-PTEN show the largest differences among the three proteins in the vicinity of the C2 domain; 8) Prior studies suggest that mutation of the five clustered Lys residues of the CBRIII loop (found on the membrane binding surface of the C2 domain) decreases lipid binding in *in vitro* vesicle binding and activity assays and signaling activity in cell transfection experiments (22, 26). We believe that points 5,6, 7, and 8 are particularly compelling in defining the Figure 8 model as they indicate that the properties of a mutant that pops the 4p-PTEN open (based on alkaline phosphatase susceptibility and anion exchange chromatography mobility) also confer enzymatic activity with soluble substrate that is characteristic of the unphosphorylated n-PTEN. Since these CBRIII loop Lys residues mutated in the D5 4p-PTEN mutant are crucial for membrane binding, it seems likely that the structure and functional changes by mutation and phosphorylation are linked, as depicted in Figure 8.

To be clear, we agree that our data do not completely eliminate alternative, indirect mechanisms, such as the phosphorylated tail communicating less directly to the C2 domain by the phospho-tail binding elsewhere on the body of PTEN. In the absence of a high resolution X-ray structure, we think this remains a viable (though unlikely) hypothetical model that we have added to the Discussion. Such an indirect model would provide a rather convoluted explanation for points 1–8 above.

Moreover, it would not easily account for why the A5 (five clustered Lys residues of the CBRIII loop replaced by Ala residues) 4p-PTEN mutant remains in the closed conformation as discussed in the manuscript. It is simple to imagine that the D5 mutant Asp residues could show electrostatic repulsion with the highly negatively charged phospho-tail whereas the 5A mutant would lack this effect. If the phospho-tail were binding elsewhere on PTEN, we would not expect the A5 and D5 clustered mutations to induce distinct conformational effects that would result in differential alkaline phosphatase susceptibility of the tail phosphates.

Moreover, in contrast to the Figure 8 model, it is within the realm of possibility that 4p-PTEN can bind to membrane in the closed conformation, albeit with significantly reduced affinity. There is no simple biophysical approach that can easily rule this out. Our best evidence against the possibility that conformationally closed 4p-PTEN binds membrane is that the interfacial *K*_m_s (i*K*_m_s) for 4p-PTEN, n-PTEN, and t-PTEN are all the same, within error, in dephosphorylating vesicle-bound PIP_3_. Since the *K*_m_s for soluble PIP_3_ dephosphorylation are different for 4p-PTEN and n-PTEN, it would be surprising that the i*K*_m_s would be the same if the conformation of 4p-PTEN were closed when bound to vesicle. It is plausible, however, that some closed 4p-PTEN with reduced membrane affinity is interacting with vesicle while in a catalytically impaired state, which would contribute only in a minor way to the i*K*_m_. On the other hand, we also know how important the CBRIII loop is for membrane binding which, as discussed above (and below), is likely abutting the phospho-tail in the closed conformation. Thus, the information available strongly supports the model in Figure 8. We discuss the alternate possibilities, and why they are disfavored in our opinion, in greater depth and clarity in the revised manuscript.

*2) Figure 2 suggests that 4p-PTEN might reduce k*_*cat*_*, as well as k*_*cat*_*/K*_*m*_*. More evidence is needed proving that the effect of phosphorylation on catalysis can be explained only by reduced phospholipid binding. If possible, k*_*cat*_
*should be measured with less error (e.g., using alternative substrates), in order to eliminate the possibility that phosphorylation affects turnover rate as well as binding constants, which would indicate mechanisms beyond simple steric interference with the active site*.

Regarding the issue of Figure 2 not ruling out a k_cat_ effect, we agree. However, we do not think that this is a key issue to the main conclusions of our manuscript or the (new) Figure 8 model. Figure 2 involves the soluble PIP_3_ substrate, rather than the more physiologic, vesicle embedded PIP_3_ form. As discussed previously, the critical micellar concentration (CMC) of diC6 PIP_3_ is greater than 1 mM, significantly higher than the concentrations used in this study (3). The mechanism of PTEN interaction with soluble substrate is at least somewhat distinct from the way PTEN interacts with vesicles, since the Lys residues of the C2 domain CBRIII loop are dispensable for efficient catalysis with soluble substrate, as first shown by Peter Downes and colleagues (26), but they are important for vesicle binding and tumor suppressive ability ([22], [26], [41], Shenoy et al. 2013). Thus, whether tail-phosphorylation of PTEN induces a pure K_m_ effect or a mixed k_cat_ and K_m_ effect with soluble PIP_3_ substrate does not impact our understanding of the mechanism by which 4p-PTEN relative to n-PTEN binds vesicles more weakly, as demonstrated in both the interfacial kinetics and the sedimentation studies. Thus, we do not see a large benefit in trying to saturate the 4p-PTEN velocity versus soluble substrate concentration plot in Figure 2. We do think that Figure 2 and soluble substrate assays add value to the manuscript because they: 1) show that the catalytic domain of PTEN is at least subtly altered in the closed conformation of 4p-PTEN relative to n-PTEN; 2) suggest that when 4p-PTEN binds vesicles, it opens, since the i*K*_m_ of 4p-PTEN matches that of n-PTEN and t-PTEN; 3) provide a convenient method for distinguishing open and closed forms that help substantiate the more open state of the D5 4p-PTEN; 4) could ultimately be used for high throughput screening for allosteric activators of 4p-PTEN.

We have also carried out an additional experiment with D5 mutant 4p-PTEN in which we analyze the concentration dependence of soluble PIP_3_ on its activity (new Figure 6). The model would suggest an intermediate K_m_ value between that measured for wild-type 4p-PTEN and n-PTEN, since 4p-PTEN is closed and n-PTEN is open, and 5D 4p-PTEN is believed to be partially open. This value, which shows an apparent soluble PIP_3_ K_m_ of 110 μM for D5 mutant 4p-PTEN, is about 2-fold greater than that of wild-type n-PTEN but less than 4p-PTEN (K_m_ >>160 μM). Thus, this new data further corroborates the model of Figure 8, and we add these results and a brief discussion on these points to the revised manuscript.

*Additional comments from Reviewer 3 based on the discussion between the reviewers: Discussion on comment 1: With respect to the presented “autoinhibitory model” it would be possible for the authors to indicate that the model that is put forward in the paper is one of many possibilities, and perhaps present hypotheses for alternate models that might be consistent with their observations*.

We have broadened this discussion as noted above.

*Discussion on comment 2: The additional work that would be needed to demonstrate autoinhibition would require (1) showing similar kinetic parameters between t-PTEN and n-PTEN, and (2) showing that the phosphorylated C-terminal peptide inhibits t-PTEN in trans down to an activity comparable to 4p-PTEN. In order to argue that autoinhibition works by interfering with lipid binding, evidence would be needed showing that (3) the phospho-C-terminal peptide interacts with the lipid-binding interface, and (4) that conservative mutations at the lipid-binding interface interfere with peptide binding and relieve autoinhibition of 4p-PTEN. Other strategies might also be suitable, but these are among the experiments that I believe were used to demonstrate autoinhibition in Src kinases*.

During the course of these studies we were somewhat reticent about pursuing the intermolecular line of investigation. We were concerned that, by disconnecting the tail from the body, there can be unnatural interactions that may not be seen with the intact molecule, where the physical attachment may help to define the orientation and energetics of the tail binding to the body. Nevertheless, the reviewers’ comments helped persuade us of the value of this complementary approach in illuminating whether the tail works as an autonomous module in regulating the body. Thus we have performed this series of experiments (i–v) as suggested and have shown: i) the lipid phosphatase kinetics with soluble PIP_3_ substrate t-PTEN are very similar to those of n-PTEN, ii) the C-tail phosphopeptide of PTEN potently inhibits t-PTEN lipid phosphatase activity with an IC50 of 1 μM, iii) the C-tail unphosphorylated peptide of PTEN has minimal effect on the enzymatic activity of t-PTEN, establishing the key role of the phosphate modifications, iv) the enzymatic activity of the D5 mutant of t-PTEN loses sensitivity to C-tail phosphopeptide, suggesting that the lipid-binding interface of the C2 domain is crucial to phosphopeptide interaction, v) addition of the C-tail phosphopeptide reduces t-PTEN binding to lipid vesicles. This additional set of experiments (shown as new Figure 7) is fully consistent with, and helps support the connection between a phosphotail-C2 interaction and the functional inhibition of enzymatic activity and lipid binding as posited in the Figure 8 model. Beyond helping to confirm the model in Figure 8, these trans tail–body experiments revealed some unexpected but interesting new aspects of the molecular recognition in phospho-PTEN. We did not anticipate the high apparent affinity of phosphopeptide tail binding to the PTEN body (ca. Kd 1 μM) based on the estimated 25:1 closed:open equilibrium measured in the alkaline phosphatase experiments with the intact 4p-PTEN. While there are several potential explanations for this high apparent affinity, one intriguing possibility is that adoption of the closed conformation of 4p-PTEN may be associated with energetic strain, a cost not incurred in the intermolecular system. Another unexpected finding was the nearly complete inhibition (>95%) of t-PTEN enzymatic activity with 10 μM phosphopeptide, exceeding the apparent rate differential (6-fold) measured with 4p-PTEN vs. n-PTEN. There may be multiple reasons for this, but it is possible that the intermolecular binding event with 4p-PTEN pieces could result from structural interactions that are not thermodynamically accessible in the intramolecular case. Regardless, these additional findings should provide us and others with important topics for future investigation.

*The question about k*_*cat*_
*could readily be addressed using p-nitrophenylphosphate or perhaps another short chain acyl PI3,4,5P3 (e.g., C2, C3, C4) with greater solubility in aqueous solution*.

It is possible that saturation with 4p-PTEN and a PIP_3_ surrogate could be achieved. But, as mentioned above, we do not think this is central to the main conclusions of the manuscript, or the model in Figure 8. While understanding how phosphorylation of PTEN affects the specific binding interaction of a soluble substrate to the PTEN active site may have some value, it is likely at least somewhat different from how the enzyme interacts with vesicle embedded phospholipid substrate. We believe that our original results and additional findings in the revised manuscript clearly demonstrate that the K_m_ for soluble PIP_3_ is increased by phosphorylation (and intramolecular conformational change) of PTEN, and this result is noteworthy regardless of whether k_cat_ is also affected.